# Luminescence Properties of Epitaxial Cu_2_O Thin Films Electrodeposited on Metallic Substrates and Cu_2_O Single Crystals

**DOI:** 10.3390/ma16124349

**Published:** 2023-06-13

**Authors:** Laima Trinkler, Dajin Dai, Liuwen Chang, Mitch Ming-Chi Chou, Tzu-Ying Wu, Jevgenijs Gabrusenoks, Dace Nilova, Rihards Ruska, Baiba Berzina, Ramunas Nedzinskas

**Affiliations:** 1Institute of Solid State Physics, University of Latvia, Kengaraga St. 8, LV-1063 Riga, Latvia; jevgenijs.gabrusenoks@cfi.lu.lv (J.G.); dace.nilova@cfi.lu.lv (D.N.); rihards.ruska@cfi.lu.lv (R.R.); baiba.berzina@cfi.lu.lv (B.B.); ramunas.nedzinskas@ftmc.lt (R.N.); 2Department of Materials and Optoelectronic Science, National Sun Yat-sen University, Kaohsiung 80424, Taiwan; ailiuioi500@gmail.com (D.D.); lwchang@mail.nsysu.edu.tw (L.C.); mitch@mail.nsysu.edu.tw (M.M.-C.C.); t0983263595@gmail.com (T.-Y.W.); 3Center of Crystal Research, National Sun Yat-sen University, Kaohsiung 80424, Taiwan; 4Center for Physical Sciences and Technology, Saulėtekio Ave. 3, LT-10257 Vilnius, Lithuania

**Keywords:** cuprous oxide, single crystal, epitaxial thin films, electrodeposition, photoluminescence, optical absorption

## Abstract

The luminescent properties of epitaxial Cu_2_O thin films were studied in 10–300 K temperature range and compared with the luminescent properties of Cu_2_O single crystals. Cu_2_O thin films were deposited epitaxially via the electrodeposition method on either Cu or Ag substrates at different processing parameters, which determined the epitaxial orientation relationships. Cu_2_O (100) and (111) single crystal samples were cut from a crystal rod grown using the floating zone method. Luminescence spectra of thin films contain the same emission bands as single crystals around 720, 810 and 910 nm, characterizing $V_{O}^{2+}$, $V_{O}^{+}$ and $V_{\mathrm{Cu}}$ defects, correspondingly. Additional emission bands, whose origin is under discussion, are observed around 650–680 nm, while the exciton features are negligibly small. The relative mutual contribution of the emission bands varies depending on the thin film sample. The existence of the domains of crystallites with different orientations determines the polarization of luminescence. The PL of both Cu_2_O thin films and single crystals is characterized by negative thermal quenching in the low-temperature region; the reason of this phenomenon is discussed.

## 1. Introduction

Cuprous oxide Cu_2_O is known as a *p*-type semiconductor material with a direct band gap of around 2.1 eV. Cu_2_O crystal has a cubic structure of the space group *Pn3m*; the primitive cell contains 2 Cu_2_O units characterized with a lattice parameter 4.27 Å [1]. The *p*-type conductivity of this semiconductor oxide is caused by natural cation deficiency, determining the presence of copper vacancies $V_{\mathrm{Cu}}$, which effectively provide holes due to the small acceptor ionization energy (i.e., shallow acceptor level) [2,3]. Due to their electronic structure and uppermost properties high-quality Cu_2_O single crystals are used as model crystals for basic studies of excitons. Historically this material was famous for fundamental investigations of hydrogen-like exciton series, their properties and Bose–Einstein condensation [4,5,6]. The recent discovery of giant Rydberg excitons [7,8]—large optical nonlinearities with ultra-strong mutual interactions and high sensitivity to external fields—has opened prospects for new applications in quantum sensing and nonlinear optics at the single-photon level. Another practical application of Cu_2_O single crystal wafer could be a substrate for growing ZnO-MgO epilayers [9] due to the low lattice mismatch (lattice constant 0.427 nm of Cu_2_O versus 0.421–0.426 nm for ZnO-MgO).

At present this material attracts great interest for its potential applications in solar energy conversion and photocatalysis at a low production cost [10] due to its favorable qualities, such as high absorption coefficient, suitable band gap width, non-toxicity and abundant reserves on Earth. Additionally, the tailored architectures of Cu_2_O crystals are at the top of researchers’ interest enabling the development of Cu_2_O with controllable size, shape, facet, defect, dopant and heterostructure [11]. Cu_2_O-based thin films and heterostructures are the most suitable forms of this material for use in solar cells, electrochromic devices, photocatalytic and antimicrobial applications. Additionally, they can be widely used in various devices, such as thin film transistors, smart windows, IR detectors, optical limiters, spintronics, gas and glucose sensing and others [12,13,14]. Possibilities of controllable variation of the thin film fabrication conditions and doping allow to attain the required functional properties.

To implement the application tasks, the mass production of high-quality Cu_2_O thin films with reproducible properties is necessary. Various methods have been used to deposit Cu_2_O thin films, such as thermal oxidation, reactive sputtering, chemical vapor deposition (CVD), pulse laser deposition (PLD), molecular beam epitaxy (MBE) and electrodeposition [15,16,17,18,19,20,21,22,23]. Except a few cases of films grown using epitaxial techniques, such as CVD, PLD and MBE, Cu_2_O thin films prepared via thermal oxidation, sputtering and electrodeposition are typically polycrystalline containing grains of 50 to 500 nm in size. The electrical and optical properties of the thin films are therefore strongly influenced by the production method which determines the microstructure and defect types as well as their amount. [5,13,24,25,26,27,28]. It is known that abundant localized states are present at grain boundaries. The undesirable states can be eliminated by growing thin films in an epitaxial form. However, the above-mentioned techniques for epitaxial growth require high temperature and high facility expense [18,19,20,21]. One exception is that the epitaxy of Cu_2_O has been realized via electrodeposition [29]. As many as five orientation relationships between Cu_2_O and Ag substrate were recently identified by employing a combinatorial substrate approach and epilayers containing low dislocation densities were obtained [30].

In-depth investigation of the photoluminescence (PL) characteristics has been carried out for thermally oxidized and sputter-deposited Cu_2_O [31,32,33]. Broad peaks at 1.72 eV for $V_{O}^{2+}$, 1.53 eV for $V_{O}^{+}$, and 1.35 eV for $V_{\mathrm{Cu}}$ were observed without band-to-band transition peaks at an energy of 1.9 eV or higher. However, detailed analyses for the electrodeposited Cu_2_O thin films are seldom reported [27]. In addition to the microstructural defects resulting from the thin film deposition, the photovoltaic and photocatalytic properties of Cu_2_O are also strongly affected by the films’ orientations or exposed surface facets. The electrical conductivity, photovoltaics, photocatalysis and thermoelectricity properties of Cu_2_O are all found to be facet-dependent [34,35,36]. Under these circumstances, it is worth examining whether the anisotropic facet effect is present in the photoluminescence characteristics. The present study therefore deals with the spectral characterization of Cu_2_O single crystals and epitaxial thin films produced via the electrodeposition method, of various orientations and exposed facets. Results indicate that the PL properties of Cu_2_O are dominated by process-related defects rather than facets. 

## 2. Materials and Methods

Cu_2_O single crystal bar of 7 mm in diameter was grown using the floating zone method. Two samples of 1 mm in thickness, with (100) and (111) polished surfaces, respectively, were cut from the single crystal bar. The single crystals are designated as sample SC(100) and SC(111), respectively, hereafter.

Five thin film samples were prepared via the electrodeposition method in aqueous solutions containing 0.4 M cupric sulfate (CuSO_4_·5H_2_O, 99.5% purity) and lactic acid (CH_3_CH(OH)COOH, 90% purity). The pretreatment of the Ag and Cu substrates were reported previously [30,37]. The electrolyte was adjusted to a pH value of 9 or 12 by sodium hydroxide (NaOH, 99% purity). The electrodeposition was carried out in a three-electrode system using a Metrohm Autolab PGSTA204 potentiostat in either a potentiostatic or galvanostatic condition at 60 °C. The electrolyte and deposition conditions of the Cu_2_O samples are listed in Table 1. The crystal structures and texture of the deposited films were characterized via X-ray diffraction (Bruker D2 X-ray diffractometer with Cu Kα radiation) in a locked-coupled (*θ*-2*θ*) mode. The (111) epilayer was analyzed using a four-circle diffractometer (Bruker D8 Discover, Cu Kα radiation) equipped with a Göbel mirror and a four-bounce monochromator. In addition to ω-2θ diffraction pattern, rocking curve (*ω* scan) of the (111)_Cu2O_ peak as well as the phi (*φ*) scans of the (200)_Cu_ and (200)_Cu2O_ peaks were also acquired for the epilayer sample (sample 111). The surface morphology of the films was observed via secondary electron microscopy (SEM, Zeiss Supra 55). As will be demonstrated later, these five samples have different orientation/facet combinations, allowing the anisotropy effect on PL property to be examined in detail. Samples 13, 30, 39 and 40 have Ag polycrystalline substrate, and sample 111 has a (111) Cu single-crystal substrate.

Optical absorption spectra were measured using a spectrophotometer Specord 210 (AnalytikJena). Cu_2_O single crystal samples were put in a closed cycle refrigerator (CCS-100/204, Janis Research Corporation, Woburn, MA,USA), providing 10–300 K temperature range, which in its turn was inserted into the spectrophotometer sample camera. Ca_2_O thin film samples on metal substrates are nontransparent, so they cannot be measured in the transmission mode. Instead, the diffuse reflection mode was used, inserting the integrating sphere with a thin film sample attached into the spectrophotometer’s sample camera. Then, the spectrophotometer’s software was used to convert reflection data into absorption spectra. These measurements were implemented at room temperature.

Photoluminescence spectra (PL) were measured under excitation with a 532 nm laser line using two luminescence-recording systems. Thin films were studied using a set-up containing a grating monochromator (Andor Shamrock SR-303i-B) together with the CCD camera (Andor DV 420A-BU2). The laser excitation light of the laser diode 532 nm (Class IIIb) was focused on a spot around 3 mm^2^. This luminescence set-up was also used for studying luminescence polarization. For that purpose, a wire grid polarizer (WP 25M—UB, Thorlabs), operating as a polarization analyzer, was inserted in the luminescence beam and a quartz-wedge achromatic depolarizer (DPU-25, Thorlabs) was located before the slit of the polarization-sensitive grating spectrometer for depolarization of the entering light. The laser, whose emission light is polarized, was positioned with the (electrical component) polarization vector vertically in relation to the sample surface. The luminescence polarization of Cu_2_O thin films was analyzed by turning the analyzer. A narrow spectral feature at 810 nm and undulations in the long wavelength region, seen in all the spectra obtained on this set-up, are due to the irremovable peculiarities of the equipment.

PL spectra of the crystal samples were recorded using a triple grating Raman spectrometer (TriVista 777, Spectroscopy & imaging GmbH, Warstein, Germany) fitted with Peltier cooled CCD detector. The Nd/YAG laser radiation second harmonic was focused on the area of several microns on the sample, and luminescence light was collected using the microscope (Olympus BX53). The spectral resolution of luminescence detection was 0.1 nm. The temperature was controlled via a temperature controller (Lake Shore 331) providing a temperature stability of 0.1 K. In this set-up, the laser power was varied by two orders of magnitude to detect the spectral changes produced by the rise of the excitation power. Metal wire cloth was used in front of the luminescence detection system to prevent its oversaturation when using the high laser excitation power. Measurement temperature in the 10–300 K range was provided via a closed-cycle helium cryostat model DE-204. All necessary apparatus spectral corrections were carried out.

## 3. Results

### 3.1. Microstructure

Figure 1 shows the SEM micrographs and XRD patterns of samples 13, 30, 39 and 40, electrodeposited on Ag polycrystalline substrates. All the low-magnification SEM micrographs shown in Figure 1a,d,g,j reveal a clear grain-based contrast. In other words, the film deposited on each substrate grain has a specific brightness, indicating that the morphology of the deposited film is affected strongly by the orientation of the substrate grain. This orientation-dependent contrast proves that the films are likely deposited epitaxially [30,37]. Moreover, the SEM micrographs at high magnifications (Figure 1b,e,h,k) acquired from the squared area in the corresponding low-magnification micrographs show that the films are composed of crystals oriented in the same manner, confirming the hypothesis of epitaxy. Accordingly, most of the films were deposited epitaxially on their underlying substrate grains. The epilayer deposited on a specific substrate grain hereinafter is thus called a domain. It is known that the Cu_2_O crystals have either {111} or {100} habit planes depending on the depositing conditions. In the present case, the SEM micrographs in Figure 1e,k, for samples 30 and 40, respectively, show that the crystals are encapsulated by {100} planes. In Figure 1b,h, no clear habit planes can be identified. With the assistance of the crystals grown on other areas (results not shown), the crystals in samples 13 and 39 have {111} habit planes. In addition, the XRD diffraction patterns acquired in an *θ*-2*θ* configuration for these samples are shown in Figure 1c,f,i,l, respectively, to reveal the coherent length, strain and preferred orientation of the deposited films. The intensity ratios among peaks of Cu_2_O should be roughly the same as those of Ag if the films were all grown epitaxially based on a cube-on-cube orientation relationship (OR). However, it is worth mentioning that at least four more ORs were identified between Cu_2_O and Ag in addition to the expected cube-on-cube OR [30]. Therefore, the intensities of the Cu_2_O peaks vary from one sample to another. The grain orientation is expressed as an <*hkl*>//ND notation; here, ND represents the normal direction of the sample surface. A measure of the relative amounts of the <*hkl*>//ND texture component was obtained from the peak intensities according to Formula (1) as proposed in [38]:(1)$$f_{hkl}=\frac{I_{hkl}}{\sum I_{hkl}}$$
where, $\sum I_{hkl}$ is the summation of the intensities of five diffraction peaks, (111), (200), (211), (220) and (311). Both samples 13 and 30 possess a strong <111>//ND texture, whereas sample 39 has a <220>//ND + <200>//ND texture and sample 40, on the other hand, has a <220>//ND + <111>//ND texture. The texture and crystal facet planes for these samples are given in Table 2. In addition, the strain and coherently scattering domain size of the four samples (13, 30, 39 and 40) are analyzed using the Williamson–Hall method [39]. The instrumental peak broadening is calibrated from the silver peaks. Samples 12 and 30 have relatively low strain values of 0.2% and 0.3%, respectively, whereas those of samples 39 and 40 are 1.2%, indicating that samples 39 and 40 have a high dislocation density as compared to that of samples 13 and 30. Samples 13 and 30 have average domain sizes of 37 nm and 46 nm, respectively. The domain sizes of sample 39 and 40 are not detectable since the intercepts of the Williamson–Hall plot are negative.

Figure 2 shows the XRD analytical results for sample 111, for which a (111) Cu single crystal substrate was used. The diffraction pattern in Figure 2 acquired in a ω/2θ configuration shows a (111)_Cu2O_ peak at 36.62° and a very weak (200)_Cu2O_ peak at 42.39° in addition to a strong (111)_Cu_ peak, indicating that an (111) epilayer was deposited. According to the (111)_Cu2O_ rocking curve in the inset, the peak has a full width at half maximum (FWHM) of 1.36°. Considering that the FWHM of the (111)_Cu_ rocking curve is about 0.3°, the crystallinity of the epilayer is only slightly worse than those prepared via the high vacuum, high temperature processes, such as molecular beam epitaxy (FWHM = 0.52° in [40]). Figure 2b shows the φ scans of the (200)_Cu_ and (200)_Cu2O_ peaks, revealing that the epilayer contains a small fraction of a <111>/60° twin variant. The SEM micrograph in the inset shows that the twin crystals can be identified in the epilayer as highlighted by the dashed triangles. Scattered crystals in a square shape enclosed by four {111} habit planes were observed occasionally on the surface of the epilayer (result not shown). These crystals have a <100>//ND orientation, corresponding well with the very weak peak present in Figure 2a at 2θ = 42.3°. Therefore, the (111) epilayer exhibits mainly a smooth (111) surface.

### 3.2. Absorption

Absorption spectra were measured to estimate the optical band gap values for the Cu_2_O samples. Absorption spectra of Cu_2_O single crystals show a band edge around 600 nm (2.07 eV) at 10 K, which is sequentially moved towards lower energies with temperature rise up to 640 nm (1.94 eV) at RT, see Figure 3a,b. Elements of fine structure are seen in the absorption spectra of Cu_2_O crystals at 582 nm (2.13 eV) (SC(111) and SC(100) samples) and 548 nm (2.26 eV) (sample SC(111)) at the lowest temperature.

For Cu_2_O film samples, due to the small thickness of the Cu_2_O film, the absorption spectrum obtained through the procedure described above (in Section 2), contained features of both the Cu_2_O layer and the substrate metal, constituting a combined spectrum. To obtain the untroubled Cu_2_O thin films absorption spectrum, the absorption spectrum of a metal substrate was measured and subtracted from the combined spectrum. As a result of this procedure, absorption spectra of Cu_2_O thin films were obtained with a large amount of uncertainty. Examples of such residual spectra are shown in Figure 3c,d for samples 13 and 39, respectively.

The widely used Tauc procedure was applied for the calculations of the band gap value *E_g_*. The band gap values were determined for all the studied Cu_2_O samples, applying the Tauc plot to the absorption curves. In the case of direct electron transitions characteristic for Cu_2_O, the photon energy and absorption coefficient are linked to band gap energy *E_g_* by relation (2) [41]:(2)$$\alpha h\upsilon =K{\left(h\upsilon -E_{g}\right)}^{\frac{1}{2}}$$
where, *h* is the Planck constant, *ν* is photon’s frequency, *E_g_* is bandgap energy, *K* is a constant, and *α* is the absorption coefficient. From the Tauc plot, the band gap value was determined as an intersection point of the tangent to a curve with an abscissa, see Appendix A for single crystals and Appendix A for thin films. The obtained band gap values for all Cu_2_O samples are shown in Table 3.

### 3.3. Photoluminescence

#### 3.3.1. Cu_2_O Single Crystals

Photoluminescence measurements were performed for two single-crystal samples of Cu_2_O, cut in (111) and (100) directions from the same crystal rod. PL excitation was carried out by the 532 nm laser applying two values of radiation power, which varied by about two orders of magnitude: 0.2 and 15 mW. The PL spectra for two single crystal samples are shown in Figure 4, corresponding to the low laser power on the top figures and the high laser power on the bottom figures. The luminescence bands observed are related to the following defects: 609 nm (2.04 eV) with an exciton, 720 nm (1.72 eV) with a doubly charged oxygen vacancy $V_{O}^{2+}$, 812 nm (1.53 eV) with an oxygen vacancy $V_{O}^{+}$, and 910 nm (1.36 eV) with a copper vacancy $V_{\mathrm{Cu}}$. (The peak positions hereafter will be titled as detected at 10 K temperature). Similar emission bands were observed and assigned by other authors [42,43,44] with a reminder that the exact peak position and relative intensities depend on the used growth procedure and sample treatment.

Exciton band at around 610 nm assigned to the yellow 1s ortho-exciton occurring due to electric quadrupole transition and denoted as (Y 1 s) followed with phonon side bands due to the electric dipole transitions and its evolution with temperature rise has been widely studied [28,42,45]. In the present paper, we will not focus on the investigation of the exciton emission band.

The relative contribution of the bands to the PL emission spectra depends on the sample. Thus, under the lower excitation light power for sample SC(111) (Figure 4a), the 910 nm band dominates, the 720 nm band is very weak and other bands are negligibly small. At the same time, for the sample SC(100) (Figure 4c), intensity of the 720 nm band is comparable with that of the 910 nm band, while the 610 nm and 812 nm bands become observable. The increase in the excitation light power causes an essential increase in the shorter wavelength bands’ intensity for both crystal samples, as seen in Figure 4b,d.

The variations of the Cu_2_O single-crystal spectra with temperature rising up to RT were studied as follows. (1) Band positions demonstrated a red shift of 0.055 ± 0.005 eV according to the thermal narrowing of the band gap, whereas the dependence on the excitation power was not observed, see Appendix A. (2) Emission bands are subjected to thermal quenching at different rates: the 812 nm band disappears at around 100 K, the 720 nm band at around 200 K, while the 609 and 910 nm bands survive up until RT.

In both Cu_2_O single crystal samples, all the observed emission bands demonstrate a complicated behavior with temperature rise: at first, emission intensity increases, then attains plateau, and at the critical temperature *T*_0_ begins to decrease, as shown in Figure 5a. The phenomenon of luminescence intensity increasing with temperature rise is called negative thermal quenching (NTQ) [46,47], anomalous thermal quenching [48] or antiquenching [49]; the notion of critical temperature is introduced by Rechnikov [46]. It was observed that in the studied Cu_2_O single crystals value of critical temperature *T*_0_ varies in the 20–55 K limits depending on the sample, particular emission band and excitation light power.

In the present paper, the temperature dependence and the corresponding Arrhenius plot are shown graphically for the most intensive 720 and 910 nm bands of sample SC(111), see Figure 5a,b. For the 910 nm band, *T*_0_ shifts from 20 K under the higher excitation power (Figure 5a, curve 1) to 50 K under the lower excitation power (Figure 5a, curve 3), whereas for the 720 nm band, *T*_0_ varies from 35 to 55 K (Figure 5a, curves 2 and 4). At higher temperature, the corresponding curves drift together. Similar luminescence behavior is observed in sample SC(100).

Arrhenius plot (Figure 5b) enables the estimation of the activation energies *E_a_* for the 910 and 720 nm emission. Activation energy can be found from the Mott–Seitz Equation (3) [50],
(3)$$\;\;\;\;\;\;\;\;\;\;\;\;\;\;I\left(T\right)=\frac{I_{0}}{1+a\;exp\left(-\frac{E_{a}}{kT}\right)}$$
where, *I*_0_ and *I(T)* and are luminescence intensities at temperatures 0 and *T*, respectively, *k* is Boltzmann constant, and *a* is a frequency factor determined by the ratio of radiative *τ_R_* and nonradiative lifetimes *τ_NR_*: *a = τ_R_*_/_*τ_NR_*.

The obtained corresponding *E_a_* values are shown in Table 4, together with those for other samples.

Similar values of activation energies are obtained for both crystal samples: 80–90 meV for the 720 nm band and 135–150 meV for the 910 nm band. The values increase with an increase in the excitation power to 90–100 meV and 190–200 meV, correspondingly.

#### 3.3.2. Cu_2_O Thin Films

Five Cu_2_O thin film samples were studied for luminescence properties, varied for the substrate type and production conditions, as listed in Table 1. The PL spectra in the 10–300 K temperature range, their dependence on illumination spot position and luminescence polarization were examined.

For all the studied Cu_2_O thin films independently of the sample structure (texture and facet plane) the PL spectrum looks like a broad structured band in the 640–1050 nm spectral region with a peak position varying in the 650–750 nm limits. The emission subbands characteristic to bulk single crystal Cu_2_O can be distinguished; they are somewhat red-shifted (to 735, 820 and 920 nm at 10 K), diffused and overlapping. Spectral positions of the PL subbands were determined by curves deconvolution into Gaussian components, not shown graphically. Additionally, new features appear at 660 nm (in some samples), 680 and 960 nm. Exciton emission around 610 nm is not observed even at the lowest temperature. Contrary to the bulk crystalline samples, the 910 nm emission band is relatively weak.

The characteristic feature of the Cu_2_O thin films is the dependence of the PL spectra shape on the illumination spot position during the detection of the luminescence signal. Movement of the excitation spot over the surface causes changes in the spectrum peak position as well as in the luminescence intensity, as illustrated in Appendix A.

The summary of the normalized PL spectra of Cu_2_O thin films obtained under excitation 532 nm at random surface places is given in Figure 6, curves 1–5; for comparison, the PL spectra of the crystalline bulk sample (100) are also shown (Figure 6, curve 6).

Luminescence polarization measurements have shown that the shape of thin films’ PL spectra depends on the angular position of the analyzer, confirming the presence of luminescence polarization. It should be mentioned that the spectra of polarized luminescence also vary depending on the illumination spot location on the sample surface.

To characterize the thermal behavior of the thin film PL at different wavelengths, the intensity of each subband integrated over the spectral range (λ_max_ ± 10) nm was plotted versus temperature. Arrhenius plot was built, and activation energies were estimated similarly to the case of single crystals and listed in Table 4. Further result description is structured according to the individual samples.

Sample 111 is a 1.0 μm thick Cu_2_O epilayer on a substrate of single crystal Cu (111). It was observed that movement of the illumination spot along the surface of the sample 111 did not cause the observable changes in the spectrum shape. The maximum of its PL spectrum at 10 K is found at 735 nm (Figure 7a). With temperature rise, the PL intensity first increases up to 30 K (Figure 8a), then decreases up to RT. Emission subbands constituting the PL spectrum are subjected to thermal quenching with different rates.

Dependence of the integral intensity of the subbands 680, 735, 820 and 920 nm on temperature is shown in Figure 8a, while its Arrhenius plot is shown in Figure 8b. All subbands demonstrate NTQ up to *T*_0_ = 30 K. With temperature increase, the 680 nm first quenches, then the 735 nm band, and the 820 and 920 nm bands survive till RT. The descending parts of the Arrhenius curves cannot be approximated with single straight lines, making the determination of activation energy values inaccurate. Roughly estimated values of activation energies for PL subbands are listed in Table 4.

PL polarization measurements confirm the presence of polarized emission of the thin film sample 111; for the particular sample position the relative contribution of the 680, 820 and 920 nm bands is lowest at the analyzer angular degree 0°, and highest at 90° (see Figure 7b).

Sample 13 is a 8.7 μm thick Cu_2_O film on a polycrystalline Ag substrate; its texture is characterized as <111>//ND and habit plane as {111}. The PL spectrum of this sample has a peak, whose position varies in a wide spectral range depending on the chosen location of the illumination spot. For the particular chosen location, the spectrum maximum is 700 nm at 10 K (Figure 9a). As seen from Figure 10a,b, neither emission subband demonstrates NTQ, and the PL intensities decrease monotonously beginning from 10 K. After thermal quenching of the 680 and 735 nm subbands, the PL spectrum at RT contains the 820 and 920 nm subbands. The obtained activation energies for emission bands are listed in Table 4.

Polarized luminescence for this sample was measured only in the 600–800 nm spectral range (Figure 9b) because of the insufficient emission intensity at higher wavelengths. Faint signs of polarized luminescence are seen as the irregular thickening of the spectrum overlapping curve.

Sample 30 represents Cu_2_O thin films on a polycrystalline Ag substrate; its thickness is 1.3 μm, the texture is characterized with <111>//DF, and the habit plane is {100}. The PL spectrum of sample 30 at the particular chosen illumination spot position has a peak at around 735 nm and a well-expressed shoulder at 680 nm (Figure 11a). With temperature rise, the short wavelength subbands are thermally quenched one by one, and at RT, the 920 nm subband is left; additionally, a subband around 960 nm becomes visible.

All emission subbands show NTQ up to *T*_0_ = 40 K, as it is seen in Figure 12a. Arrhenius plots for the subbands intensities can be approximated with straight lines (Figure 12b) and estimated activation energies (shown in Table 4).

Polarization measurements (Figure 11b) taken for a random illumination spot on the sample surface confirms the presence of polarized luminescence, expressed mostly in the 680 and 750 nm spectral areas.

Sample 39 is a 1.3 thick Cu_2_O film on Ag polycrystalline substrate, characterized with texture <200>//ND + <220>//ND and habit plane {111}. It has well-structured subbands at 680, 735, 820 and 920 nm in the PL spectrum (Appendix A); all subbands are characterized with NTQ, critical temperature *T*_0_ being 20 K for the 680 nm and 40 K for other subbands (Appendix A). The presence of luminescence polarization is pronounced best in the spectral regions around 700, 735 and 820 nm (Appendix A).

Sample 40 is Cu_2_O thin film on Ag polycrystalline substrate; its thickness is 1.3 μm, texture is <111>//ND + <220>//ND and habit plane is {100}. PL spectrum of this sample has evident subbands at 680 and 660 nm apart from the usual subbands 735, 820 and 920 nm (Appendix A). The 660 and 680 nm subbands thermally quench monotonously and disappear below 100 K. Other subbands show NTQ, the most high-temperature one is the 920 nm subband. It is observable at RT together with the 960 nm subband (Appendix A).

## 4. Discussion

### 4.1. Absorption

Optical absorption in Cu_2_O is determined by the direct transitions from the valence band to the conduction band at the Γ-point of the Brillouin zone. Splitting of the highest valence band and the lowest conduction band results in four types of the allowed fundamental transitions named yellow, green, blue and indigo, according to the wavelengths of their spectral positions [42]. The optical band gap, determined by the yellow transitions, is mentioned at 2.17 eV [42,43]. Electron transitions are accompanied by the corresponding series of excitonic transitions, observed in the absorption spectra. Exciton absorption lines are seen even at RT in high-quality crystals, especially after crystal annealing.

Absorption spectra of single crystals (100) and (111) obtained in the present work (Figure 3a,b) clearly show the thermal shift of the band edge by 0.13 eV in the 10–300 K range and signs of lines from the yellow exciton series at the lower temperature. The optical band gap values obtained for both crystals from the Tauc plot constitutes 2.04 eV at 10 K, which is lower than most of the literature data. 

Absorption spectra of Cu_2_O thin films obtained from the diffuse reflection spectra and with subtraction of the metal substrate spectrum are not precise. They are shown here for two thin film samples for illustrative purposes (Figure 3c,d). The band gap values for Cu_2_O thin films obtained using the Tauc plot method range in 1.91–2.12 eV limits, with an error ±0.05 eV. Such band gaps are in agreement with a very broad range of values for Cu_2_O thin films, varied depending on the production method, substrate type and growth conditions, beginning from 1.92 up to 2.02 eV for samples electrodeposited on glass [27], 2.0 eV for the samples electrodeposited on Au [24], and up to 2.4 eV in the vacuum annealed magnetron sputtered thin films on glass [14,23]. Growth conditions and annealing of thin films play a decisive role in stoichiometry of crystalline structure, determining presence of lattice defects, and the band gap value [13].

### 4.2. Photoluminescence

#### 4.2.1. Single Crystals

In general, the experimental spectra of Cu_2_O single crystals, containing emission bands consisting of an exciton (609 nm; 2.04 eV), a doubly charged oxygen vacancy $V_{O}^{2+}$ (720 nm; 1.72 eV), an oxygen vacancy $V_{O}^{+}$ (812 nm; 1.53 eV), and an electrically neutral copper vacancy $V_{\mathrm{Cu}}$ (910 nm; 1.36 eV) (see Figure 4a–d) coincide well with the photoluminescence spectrum in the 600–1050 nm spectral range reported previously [42,43]. The mentioned energies are observed at low temperature (10 K), and the emission energies decrease due to the band gap narrowing with increasing temperature. Already in the 1980s and later, it was suggested that the emission bands of Cu_2_O natural and synthetic crystals in the 700–1000 nm range are connected with bound excitons localized at oxygen and copper vacancies [6,26,51]. The varied mutual relation is observed between intensities of the emission bands in samples SC(100) and SC(111), as seen in Figure 4a,b, which becomes even more pronounced under increased excitation power (Figure 4c,d). The different relative contributions of the defect-related emission bands to the PL spectrum in samples SC(100) and SC(111) cut from the same crystal rod, should be explained rather by the inhomogeneous distribution of the intrinsic defects along the crystal rod created during growth, than by the crystallographic plane of the sample surface.

The intensity of the defect bands at 720, 812 and 910 nm gives no information on the relative abundance of the corresponding defects. The corresponding luminescence centers are characterized by different physical parameters, exciton localization mechanisms and luminescence mechanisms [26]. The rise of the excitation power increases the number of excitons and conduction carriers in a crystal, localizing at oxygen and copper vacancies with different efficiency. Thus, in [28] it was supposed that conduction carriers or other nonthermalized states tend to be trapped at oxygen vacancies, while cold and diffusive excitons tend to excite copper vacancies. Figure 4 shows that the emission of the centers corresponding to oxygen vacancies (the 720 and 812 nm bands) are more sensitive to excitation power increase than copper vacancies responsible for the 910 nm band. The dependence of the defect bands’ intensity relation on excitation power confirms the concept of their origin from bound excitons.

In the present work, it was observed that all the defect emission bands of Cu_2_O crystal samples are subjected to the negative thermal quenching NTQ, where luminescence intensity increases with temperature rise in some temperature range (see Figure 5a) for the 720 and 910 nm emission bands of the crystal sample (111). Previously NTQ of emission bands was observed in different materials and explained by different mechanisms: in doped GaN by quenching of a competitive intense luminescence band or a nonradiative channel [46], in sandwiched single layer MoS_2_ by raising radiative recombination rate of the excess delocalized carriers created from the carrier hopping from the shallow defect states to the band edges [48], in CdMnTe by the participation of intermediate states of impurities in recombination processes, and in ZnO and CdMnTe by the presence of intermediate defect states and nonradiative channels [52,53]. Some authors explain NTQ by thermal activation of charge carriers and trapped excitons from shallow traps followed by their localization at radiative centers [28,49,54].

As for Cu_2_O single crystals, only some researchers have observed NTQ of one or several defect bands in their PL studies [28,55], while others reported regular continuous thermal quenching of emission bands. The latter explanation involving retrapping of the bound excitons seems to be the most appropriate for the Cu_2_O crystals. Most probably, NTQ is observed in those Cu_2_O crystals, where surface or intrinsic defects create shallow nonradiative levels for localization of bound excitons at low temperatures, easily dissociated at temperature rise. Such defects could be produced in crystals under specific growth conditions. Released excitons are retrapped at oxygen or copper vacancies, leading to the increase in emission intensity of the defect-related bands up to the point of critical temperature; their further thermal behavior follows the normal thermal quenching.

Under increased excitation power, critical temperature shifts to higher temperatures for both prominent emission bands 720 and 910 nm, see Figure 5a. The following quenching occurs at different rates but at the final stages, the corresponding curves (Figure 5a, (curves 1 and 3); (2 and 4) flow together. Thus, the 720 nm emission band reaches complete bleaching at 150 K both at low and high excitation power. The Arrhenius plot (Figure 5b) confirms the complicated character of the quenching behavior, and the values of the activation energy can be estimated with a large error (15%), see Table 4.

#### 4.2.2. Thin Films

Many authors have studied the PL spectra of the Cu_2_O thin films produced via different methods and using different substrates [10,13,23,24,25,26,27,42,56,57]. These spectra may be either with or without exciton feature; in some samples, the defect bands are well separated, while in others, they are more diffused and merged into a continuous broad band. Similar to the Cu_2_O single crystals, luminescence bands in thin films are assigned to a doubly charged oxygen vacancy $V_{O}^{2+}$, an oxygen vacancy $V_{O}^{+}$, and an electrically neutral copper vacancy $V_{\mathrm{Cu}}$. However, band positions are shifted compared to their single crystal counterparts. Compared with the PL spectra of the single crystal Cu_2_O samples, the thin film spectra are characterized by the higher relative contribution of oxygen vacancy bands. Taking into account the relatively larger contribution of the surface defects in the case of thin films containing tiny crystallites, the domination of oxygen vacancy-related radiation is in accordance with experimental microscopic observations and computer simulations using density functional theory, which have shown that oxygen vacancy is a prevalent defect on Cu_2_O surface [58]. In [57], it is stated that though the observed *p*-type conduction behavior confirms the abundance of Cu defects in both Cu_2_O systems, in the PL spectra of the thin films, the emission bands related to oxygen vacancies dominate over those related to copper vacancies. It is explained by the larger concentration of oxygen defects in thin films compared to single crystals, which appears due to film annealing in the O_2_ atmosphere. Additionally, the recombination channel via oxygen vacancies is less sensitive to structural disorder than the copper vacancy channel, and thus more efficient in the case of thin films. 

An additional band in the 650–690 nm region is often observed [10,28,42,59], which is not typical for Cu_2_O crystals and whose origin is not determined, though its connection with nitrogen-related acceptor was suggested [10,59]. In [26], the broad structureless spectrum of Cu_2_O/Au (111) thin films was observed under 532 nm; it was suggested to be related to interface plasmons, stimulated by Cu_2_O excitons, which migrate to the metal/oxide boundary. No information on the NTQ observations in thin film samples was found in the literature. 

Summarizing the results on PL characteristics under 532 nm laser excitation of the studied Cu_2_O thin films electrodeposited on metal substrates, the general similarity of their PL spectral and thermal properties is observed independently of the film orientation and facet plane and provided by the presence of intrinsic defects. The exciton band around 610 nm is not observed or is negligibly small, and the defect-related luminescence is presented with a broad structured spectrum in the 630–1050 nm spectral range with a dominating part in the 650–750 nm region, corresponding mainly to $V_{O}^{2+}$ band. The emission bands’ structure allows to distinguish the common bands related to a doubly charged oxygen vacancy $V_{O}^{2+}$, an oxygen vacancy $V_{O}^{+}$, and an electrically neutral copper vacancy $V_{\mathrm{Cu}}$.Besides that, the additional bands are observed at 680 nm (1.82 eV) and 660 nm (1.88 eV), especially well pronounced in samples 13, 30 and 40 (Figure 6).

Most of the thin film samples, except sample 13, demonstrate NTQ in the low temperature region, critical temperature *T*_0_ occurring at 30–40 K. It can be supposed that NTQ in Cu_2_O thin films has the same origin as that in single crystals and relates to excitons delocalization from shallow trap levels and localization at oxygen and copper vacancies at temperature rise. During regular thermal quenching, due to temperature rise emission subbands disappear one after another beginning with the shortest wavelengths and ending with the longest wavelengths, so mainly 900–1000 nm emission is left at RT. From the Arrhenius plots for the studied samples, which are not described by single straight lines for most of the studied thin films, it follows that the thermal quenching is governed by a combined process, and only a rough estimation of the activation energies can be performed, especially for the $V_{\mathrm{Cu}}$ band at 920 nm. Values of activation energy for the thin films are given in Table 4.

Cu_2_O thin film layers (13, 30, 39, and 40) on Ag polycrystalline substrates demonstrate dramatic changes in spectrum shape with a shift of the excitation spot and polarized luminescence mainly in the 650–850 nm region. Dependence of the PL spectra shape on the excitation spot on a sample surface and luminescence polarization features are in accordance with the domain structure of Cu_2_O thin films, which is seen in the SEM micrographs photos (Figure 1). Uniformly oriented Cu_2_O crystallites of few microns size of cubic and pyramidal shape are arranged in domains of tens micrometer size. An excitation spot of ≈3 mm^2^ covers several crystalline domains. Therefore, spectrum shape and polarization of the detected luminescence signal is determined by the radiative defects in the irradiated domains. It has been demonstrated in [30] that each domain in a Cu_2_O sample contains grown-in dislocations of different densities. The densities of radiative defects may also be different in domains. It explains why the PL spectra shape varies from one position to another. These facts allow the proposal of chaotic distribution of domains with crystallites of different orientations and different mutual relations of luminescence bands. Irregular structure of polycrystalline metal substrate determines the irregularities of the Cu_2_O crystalline structure.

Sample 111 shows the minimal dependence of the spectrum shape on the shift of the excitation spot and at the same time the most expressive polarization features along the whole spectrum (Figure 7b), speaking in favor of uniform distribution of similar crystallites along the surface of the sample. Evidently, the regular crystal structure of Cu crystal determines the regularity of Cu_2_O crystallites.

Properties of the 660 and 680 nm emission bands, seen in the PL spectra of Cu_2_O thin films, manifested in thermal behavior and luminescence polarization, show their similarity to the $V_{O}^{2+}$ emission band at 735 nm. It implies that the 660 and 680 nm emission bands may be related to bound excitons localized at oxygen defects distorted by crystallite surface, grain boundaries, and dislocations or other extended defects, generating local electric fields and being produced during thin film growth. Similarly, the long wavelength band at around 960 nm, seen in the Cu_2_O thin films at RT, might be related to distorted copper vacancies, though additional studies are required for its assignment.

## 5. Conclusions

Spectral properties of Cu_2_O (100) and (111) single crystals, grown using the floating method, and Cu_2_O thin films, deposited via electrodeposition, with various orientations and facets were studied under laser excitation 532 nm in the 10–300 K temperature range. The optical band gap value of the Cu_2_O single crystal samples was estimated from the absorption spectrum as 2.04 eV. Similar to other reported data, the PL spectra of the studied Cu_2_O single crystal samples contain the Y1 exciton emission band (609 nm; 2.04 eV) with phonon sidebands, and emission bands related to a doubly charged oxygen vacancy $V_{O}^{2+}$ (720 nm; 1.72 eV), an oxygen vacancy $V_{O}^{+}$ (812 nm; 1.53 eV), and an electrically neutral copper vacancy $V_{\mathrm{Cu}}$ (910 nm; 1.36 eV). The negative thermal quenching of defect-related emission bands observed in the low-temperature region (up to critical temperature *T*_0_ 20–50 K) is presumably explained by the release of bound excitons from the shallow traps due to lattice structure defects generated during crystal growth and retrapping of bound excitons to oxygen and copper vacancies. The presence of NTQ confirms the concept of the defect bands origin related to bound excitons. Novel results related to the PL properties of Cu_2_O single crystals refers to the effect of the excitation light power: an increase in excitation power by two orders of magnitude causes the non-proportional rise of the emission bands related to oxygen vacancies, a decrease in the critical temperature and an activation energy of regular thermal quenching of $V_{O}^{2+}$ and $V_{\mathrm{Cu}}$ emission bands.

For the Cu_2_O thin film samples, the rough estimation of the optical band gap showed values in the 1.91–2.12 eV region at RT. The present study confirms that the PL properties of the studied Cu_2_O thin films are determined by the presence of the intrinsic defects in crystalline lattice, and not by the sample orientation and facet plane. Contrary to single-crystal counterparts, PL spectra of the studied thin films contain no exciton band, and the defect-related region is presented with a broad structured band in the 650–1050 nm region, where in addition to the distinguished diffused $V_{O}^{2+}$ (735 nm, 1.69 eV), $V_{O}^{+}$ (820 nm, 1.51 eV) and $V_{\mathrm{Cu}}$ (920 nm, 1.35 eV) bands, additional bands are observed at 660 nm (1.88 eV) in some samples, 680 nm (1.82 eV) and 960 nm (1.3 eV). In the present work, the systematic study of the temperature behavior of emission bands in Cu_2_O thin films was undertaken for the first time. It was found that under temperature rise for most samples, NTQ is observed for the emission bands at low temperature up to *T*_0_ 30–40 K, followed by normal thermal quenching. Rising temperature causes emission fading beginning at the short wavelength side. Similarity of spectral and thermal properties of the PL of Cu_2_O thin films with those of single crystals allows their assignment to the emission of bound excitons localized at the oxygen and copper vacancies and other structural defects.

The dependence of the PL spectrum shape on the position of the excitation spot and the presence of the luminescence polarization, newly measured for such structures, is in line with the domain structure of thin films, containing crystallites of different orientations and presumably having different defects. Cu_2_O epitaxial layer on Cu single crystal (111) stays apart from other samples on polycrystalline metal substrates due to its regular structure, manifested in the polarized luminescence spectrum and spectrum independence on the excitation spot.

## Figures and Tables

**Figure 1 materials-16-04349-f001:**
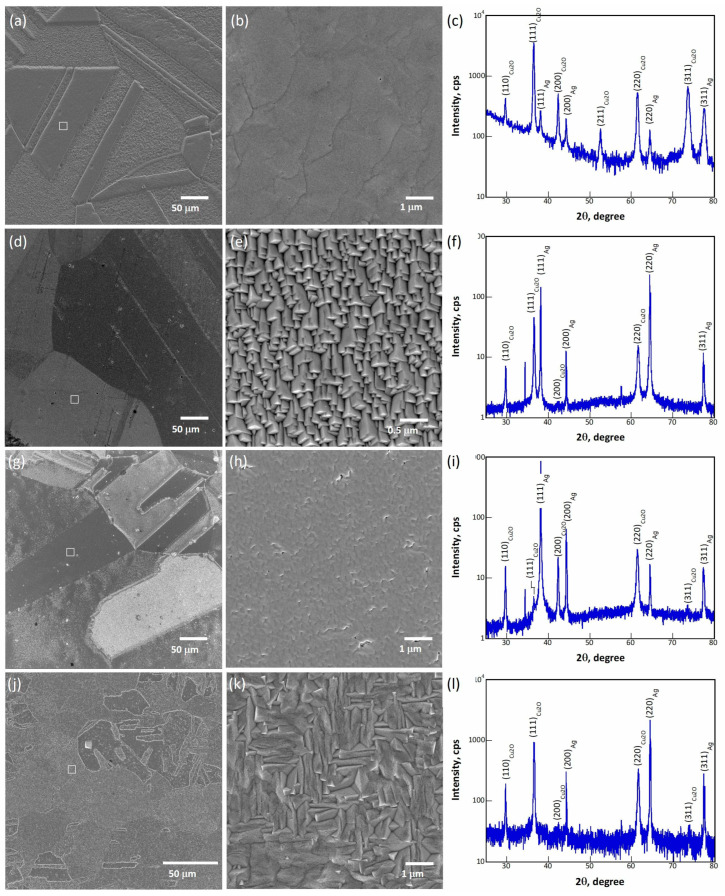
SEM micrographs of samples 13 (**a**,**b**), 30 (**d**,**e**), 39 (**g**,**h**) and 40 (**j**,**k**), and corresponding XRD patterns (13-(**c**), 30-(**f**), 39-(**i**), 40-(**l**)). The white squares in micrographs (**a**,**d**,**g**,**j**) indicate where micrographs (**b**,**e**,**h**,**k**), were acquired, respectively.

**Figure 2 materials-16-04349-f002:**
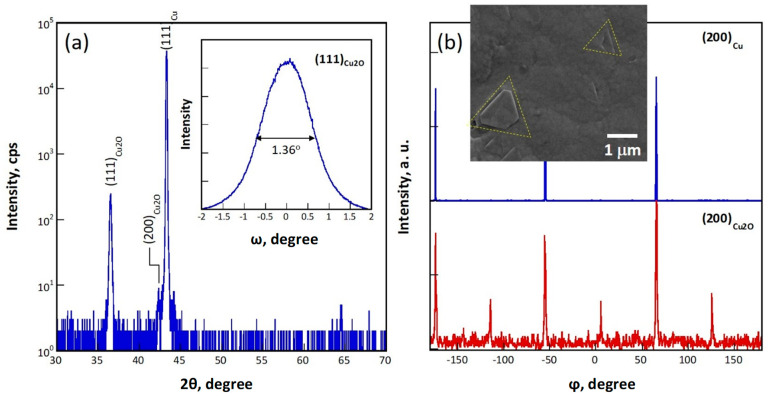
X-ray analyses for an epitaxial Cu_2_O film grown on a Cu (111) substrate (sample 111): (**a**) an ω/2θ scan with an ω scan for the (111)_Cu2O_ peak in the inset, and (**b**) φ scans of (200)_Cu_ (blue) and (200)_Cu2O_ (red) peaks with a SEM micrograph in the inset. The yellow dashed triangles in the inset SEM micrograph highlight the twinned crystals in the epitaxial film.

**Figure 3 materials-16-04349-f003:**
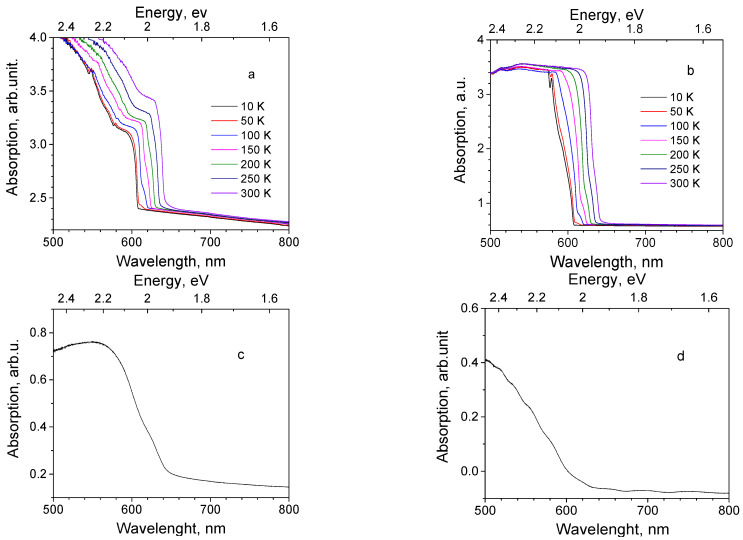
Absorption spectra of Cu_2_O samples. Single crystals at temperature, shown by legends: (**a**) sample SC(111) and (**b**) sample SC(100); thin films at RT: (**c**) sample 13, and (**d**) sample 39.

**Figure 4 materials-16-04349-f004:**
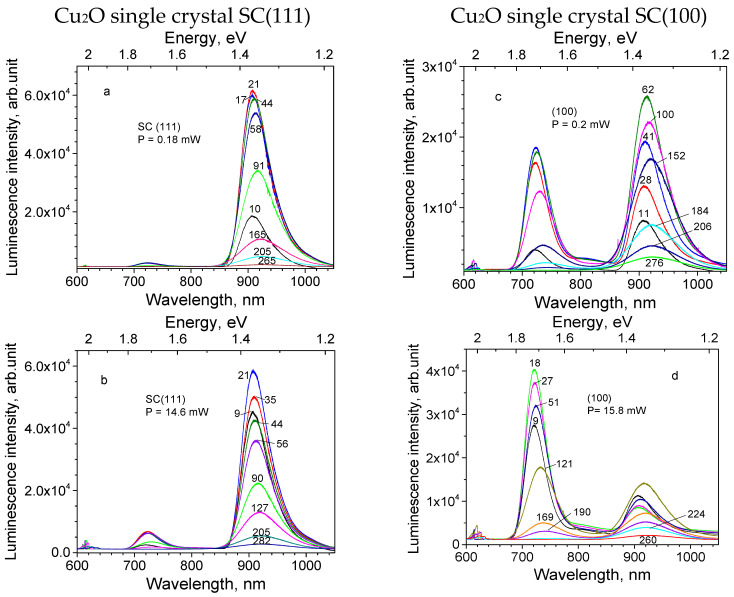
Photoluminescence of Cu_2_O single crystals under 532 nm laser excitation at varied temperatures. Sample names, applied laser power and temperature (in K) are shown on graphs.

**Figure 5 materials-16-04349-f005:**
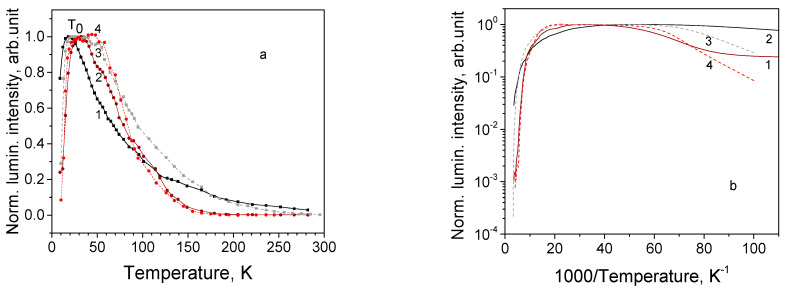
(**a**) Thermal dependence of luminescence intensity for sample SC(111) for bands and excitation powers: 1—910 nm, 15 mW; 2—720 nm, 15 mW; 3—910 nm, 0.2 mW; and 4—720 nm, 0.2 mW. (**b**) Arrhenius plot for thermal dependence of luminescence intensity for sample SC(111) for bands and excitation powers: 1—910 nm, 15 mW; 2—720 nm, 15 mW; 3—910 nm, 0.2 mW; and 4—720 nm, 0.2 mW.

**Figure 6 materials-16-04349-f006:**
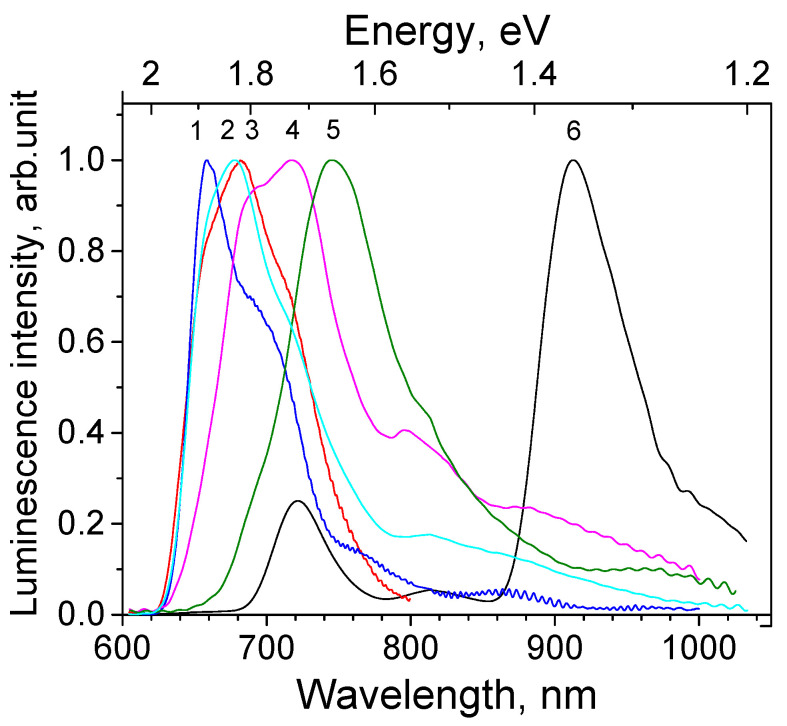
Normalized photoluminescence spectra of Cu_2_O samples under excitation with 532 nm laser at 10 K: thin films 1—30, 2—40, 3—13, 4—39, 5—111, and 6—crystalline sample SC(100).

**Figure 7 materials-16-04349-f007:**
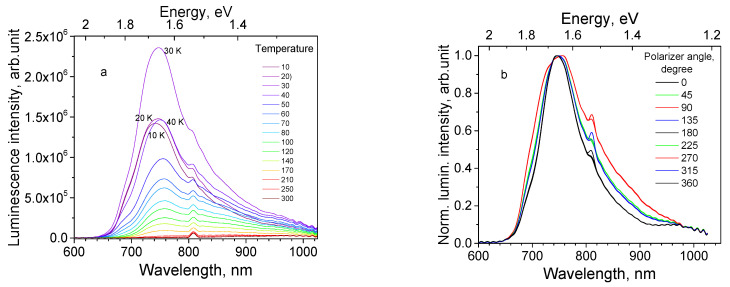
Spectral properties of Cu_2_O epilayer on Cu single crystal substrate (sample 111) under 532 nm laser excitation: (**a**) Thermal evolution of PL spectra (temperature values are shown on the graph); and (**b**) Polarized PL spectra at different angular positions of the analyzer at 10 K (angular degrees are shown on the graph).

**Figure 8 materials-16-04349-f008:**
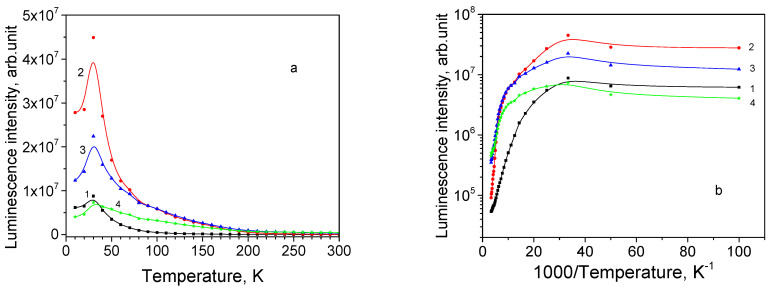
Thermal evolution of PL subbands for thin film sample 111: (**a**) PL intensity; (**b**) Arrhenius plot for the subbands: 1—680 nm, 2—735 nm, 3—820 nm, and 4—920 nm.

**Figure 9 materials-16-04349-f009:**
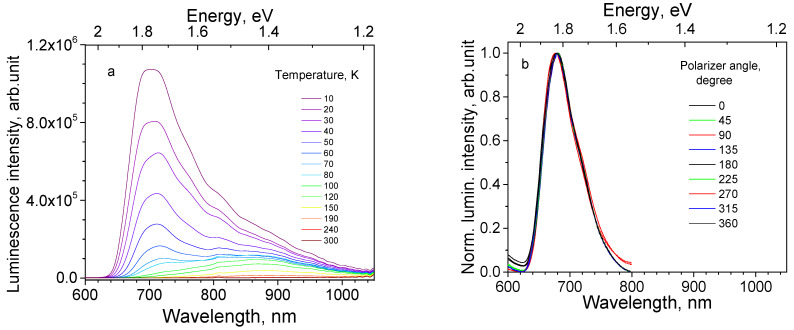
Spectral properties of Cu_2_O thin film on Ag polycrystalline substrate (sample 13) under 532 nm laser excitation: (**a**) Thermal evolution of PL spectra (temperature values are shown on the graph); and (**b**) Polarized PL spectra at different angular positions of the analyzer at 10 K (angular degrees are shown on the graph).

**Figure 10 materials-16-04349-f010:**
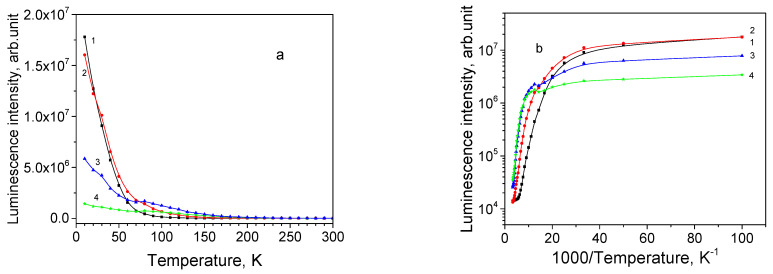
Thermal evolution of PL subbands for thin film sample 13: (**a**) PL intensity; (**b**) Arrhenius plot; 1—680 nm, 2—735 nm, 3—820 nm, and 4—920 nm.

**Figure 11 materials-16-04349-f011:**
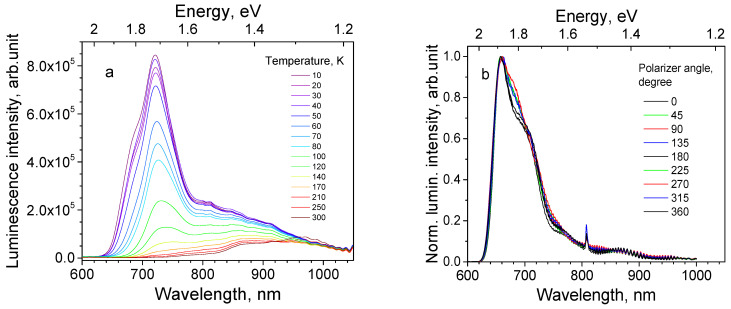
Spectral properties of Cu_2_O thin film on Ag polycrystalline substrate (sample 30) under 532 nm laser excitation: (**a**) Thermal evolution of PL spectra (temperature values are shown on the graph); (**b**) Polarized PL spectra at different angular positions of the analyzer at 10 K (angular degrees are shown on the graph).

**Figure 12 materials-16-04349-f012:**
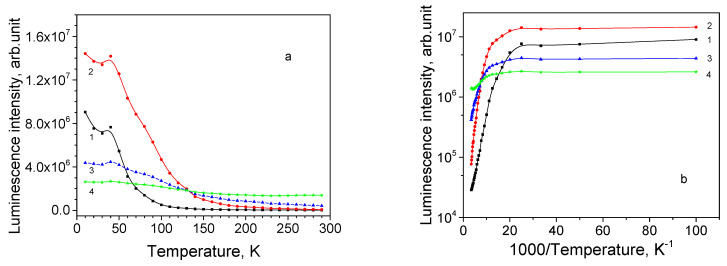
Thermal evolution of PL subbands for thin film sample 30: (**a**) PL intensity; (**b**) Arrhenius plot; 1—680 nm, 2—735 nm, 3—820 nm, and 4—920 nm.

**Table 1 materials-16-04349-t001:** Electrolyte and deposition conditions of the Cu_2_O thin film samples.

Sample Number	Substrate	Lactic Acid Concentration, M	pH	Thickness, μm	Current Density (G)/Potential (P)
13	PC Ag	1.3	12	8.7	−0.38 V (P)
30	PC Ag	3.0	12	1.3	−0.44 V (P)
39	PC Ag	3.0	9	1.3	−0.15 V (P)
40	PC Ag	4.0	12	1.3	−0.43 V (P)
111	(111) Cu	1.3	12	1.0	0.25 mA/cm^2^ (G)

PC: polycrystal.

**Table 2 materials-16-04349-t002:** Texture and crystal habit planes for the four samples.

Sample	*f_hkl_*					Texture	Facet Plane
	<111>	<200>	<211>	<220>	<311>		
13	0.64	0.05	0.02	0.16	0.13	<111>//ND	{111}
30	0.64	0	0	0.36	0	<111>//ND + <220>//ND	{100}
39	0.03	0.29	0	0.68	0	<200>//ND + <220>//ND	{111}
40	0.33	0	0	0.64	0	<111>//ND + <220>//ND	{100}
111						<111>//ND	{111}

**Table 3 materials-16-04349-t003:** Optical band gap values for Cu_2_O single crystal and thin film samples obtained from absorption measurements.

	Sample	E_g_, eV SC (100)	E_g_, eV SC (111)	E_g_, eV # 111	E_g_, eV # 13	E_g_, eV # 30	E_g_, eV # 39	E_g_, eV # 40
T, K								
10		2.04 ± 0.02	2.04 ± 0.02					
300		1.94 ± 0.02	1.94 ± 0.02	1.92 ± 0.05	1.99 ± 0.05	1.98 ± 0.05	2.12 ± 0.05	2.02 ± 0.05

**Table 4 materials-16-04349-t004:** Activation energies of subbands detected in Cu_2_O single-crystal samples (obtained with an error of ≈15%).

Sample		*E_a_*, eV 660 nm	*E*_a_, eV 680 nm	*E_a_*, eV 720 nm	*E_a_*, eV 810 nm	*E_a_*, eV 920 nm
Cu_2_O single crystal @ laser irradiation power	SC (111) @ 0.2 mW			80		150
	SC (111) @ 15 mW			93		190
	SC (100) @ 0.2 mW			89	22	135
	SC (100) @ 15 mW			97	20	197
Cu_2_O thin films	111		32	77	42	15 + 37
	13		36	43	62	65
	30		41	59	33	22
	39		59	86	37	12 + 126
	40	32	40	55	37	18 + 247

## Data Availability

The raw/processed data required to reproduce these findings cannot be shared at this time as the data also form a part of an ongoing study.

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
