# Peer review of "Luminescence Properties of Epitaxial Cu2O Thin Films Electrodeposited on Metallic Substrates and Cu2O Single Crystals"

_materials, 2023, doi:10.3390/ma16124349_

Round 1
Reviewer 1 Report
This manuscript is focused on the luminescent properties of Cu2O crystalls and thin films. The research results are original and relevant. I think this manuscript is useful to the community and acceptable for publication after some minor revisions and following issues being addressed.
1) Why was this particular thickness of the researching films chosen?
2) In conclusion, it should be add the discussion for using the material both in the form of films and in the form of single crystals. Explain their advantages.
There are some writing errors in this paper.
Author Response
Response to the Reviewer 1 is described in the file Answers to Reviewer 1

Reviewer 2 Report
The manuscript is well written, experiments and results are clearly presented and discussed in detail. There are few comments that could help the authors to further improve the paper.
Abstract, Supplementary Materials, Funding
Note the superscript and subscript of indices and exponents in the chemical formulas and when specifying the lattice defects. Replace PL with photoluminescence when using it for the first time.
Introduction
Lines 34-36: The sentence “The p-type conductivity … negatively charged copper vacancies … which are electron acceptors” needs revision/clarification.
Materials and Methods
Line 90: Add the method used (diffuse reflection) to make it clearer.
Results
Line 169: There is no Figure 2(a). The information in the caption reads (left) and (right).
Table 3: The two numbers 10 and 300 listed in the first column are not explained. What are these values?
Lines 303, 304: The temperature dependence of the PL is shown in Figure 8 (a). Since the sentence refers to this figure, this should be stated.
Author Response
Response to the reviewer is described in file Answer to Reviewer 2.

Reviewer 3 Report
The text discusses the luminescent properties of Cu2O (copper oxide) thin films and single crystals. The study includes the deposition of Cu2O thin films epitaxially using the electrodeposition method on Cu or Ag substrates at different processing parameters and comparing the luminescent properties of these thin films with those of Cu2O single crystals. The study examines the spectral properties of Cu2O samples under laser excitation in the 10-300 K temperature range, focusing on the emission bands related to various defects. The negative thermal quenching of defect-related emission bands observed in the low-temperature region is discussed, and the origin of the emission bands is explored. The text also covers the optical band gap value of the Cu2O samples, the polarization of luminescence, and the dependence of the PL spectrum shape on the position of the excitation spot. Overall, the text presents a detailed study of the luminescent properties of Cu2O thin films and single crystals, highlighting their defect-related emission bands and their behavior under different conditions.
After careful review, we have determined that your work shows promise but requires significant revisions before it can be considered for publication.
I have identified several major issues that need to be addressed before we can move forward with publication.
Firstly, we recommend that you provide a more detailed description of the methodology used in your study. While you have given an overview of the techniques used, the specifics of your experimental procedures are not clearly outlined. This would allow other researchers to replicate your work.
Secondly, we suggest that you provide additional data to support your findings. While the results you presented are intriguing, they are only convincing with more substantial evidence. Including additional experiments or analyses would significantly enhance the credibility of your work. Also, statistical analysis is required. There are no indications of the reproducibility of the optical properties of the film.
Finally, we recommend that you revise your writing for clarity and organization. For example, some of the sections of your paper could be better structured, and your writing could benefit from more precise language.
Here are my specific comments.
1. The authors make a determination of crystallinity from the determination of the FWHM of some diffraction peaks. However, there are no indications that the pattern has been refined. Then, the FWHM gets information about instrument width, grain size, and strain.
2. In that same sense, how did you determine the texture of the crystal? They should specifically show where the texture effects are located.
3. Some works like:
https://www.sciencedirect.com/science/article/pii/S0030402617317539#bib0160
https://link.springer.com/article/10.1007/s10854-016-6072-2
https://pubs.acs.org/doi/pdf/10.1021/jp061835k ,
use a different coefficient for the Tauc plot for the CU2O case. This can confuse in the way the authors have written it.
Optical absorption coefficient α of a semiconductor is energy (hv) dependent, according to the following equation:
![]()
where h is the Planck constant,
is photon’s frequency,
is bandgap energy, C is a constant, and
is the absorption coefficient which describes how much light of a given colour is absorbed by a material of given thickness. The n is a factor which depends on the nature of the electron transition and is numerically equal to 1/2, 3/2, 2 or 3 for direct allowed, direct forbidden, indirect allowed, or indirect forbidden transitions, respectively. Please rewrite the Tauc equation in its conventional form to see if you are correctly using the coefficient for direct transitions.
4. It is necessary to show the Tauc graph related to the previous question. Suppose the authors do not wish to add it directly to the manuscript. In that case, it should be added as appendices or supplementary material to determine if the Eg adjustment is appropriate.
5. It is observed that some bands shift more than others due to the effect of temperature. This may be due to a significant contribution from the surface. The authors must indicate which photoluminescence bands correspond to bulk and surface states. It is also recommended to make a graph showing how the luminance bands shift to determine if it is due to an effect of thermal expansion/compression or if there are kinetic effects of mobility of defects and vacancies due to temperature.
6. The authors must refine the diffraction patterns and analyze parameters such as crystallite size and strain.Please standardize the letter size of the legends in the figures, and use use greek letters (two thera -> 2
)
7. The correct way to refer to arbitrary units is arb. unit. Review the suggested document. Authors must correct the notation in all their graphs. Olesen, H. (1995). "Properties and Units in the Clinical Laboratory Sciences: I. Syntax and Semantic Rules (IUPAC-IFCC Recommendations 1995)" (PDF). Pure Appl. Chem. IUPAC. 87 (8/9): 1563–1574. doi:10.1351/pac199567081563
8. The graphs lack adequate identification of the curves. Since they do not have consistent and proper terminology, they are challenging to interpret. Authors must indicate what they mean given one of the lines and colors of the graphs.

Author Response
Response to the Reviewer is described in file Answer to reviewer 3.

Reviewer 4 Report
Dear Editor,
Concerning the Manuscript No.: materials-2385889, entitled “Luminescence properties of epitaxial Cu2O thin films electrodeposited on metallic substrates and Cu2O single crystals". Authors: Laima Trinkler et al.
The authors prepared Cu2O thin films epitaxial by the electrodeposition method on Cu or Ag substrates. The following major comments:
1. The subscript of the number 2 must be performed in “Cu2O” in the whole manuscript.
1. Line 42, what did you mean by [6-7 and references therein].
2. More focus on the novelty of the work must be added
3. Why didn’t the authors fix the pH value?
4. The XRD parameters must be included in a Table with suitable refinement fit.
5. The energy band gap must be obtained from Tauc plots and not from the absorption edge (inaccurate technique)
6. The PL peaks must be deconvoluted
7. The plotting skill must be enhanced
8. The FT-IR analysis must be added
9. The English must be improved.
The English must be improved
Author Response
Response to the reviewer is described in the file Answer to Reviewer 4.

Round 2
Reviewer 4 Report
The authors performed all suggestions successfully.
The English may be improved